# ComplexityMeasures.jl: Scalable software to unify and accelerate entropy and complexity timeseries analysis

**George Datseris**[1]*, **Kristian Agasøster Haaga**[2,3,4]

**1** Department of Mathematics and Statistics, University of Exeter, Exeter, United Kingdom, **2** Department of Earth Science, University of Bergen, Bergen, Norway, **3** Center for Deep Sea Research, University of Bergen, Bergen, Norway, **4** Bjerknes Centre for Climate Research, Bergen, Norway

\* g.datseris@exeter.ac.uk

**Data availability statement:** The figures and the performance numbers quoted in the comparison table are fully reproducible.

## Abstract

In the nonlinear timeseries analysis literature, countless quantities have been presented as new "entropy" or "complexity" measures, often with similar roles. The ever-increasing pool of such measures makes creating a sustainable and all-encompassing software for them difficult both conceptually and pragmatically. Such a software however would be an important tool that can aid researchers make an informed decision of which measure to use and for which application, as well as accelerate novel research. Here we present ComplexityMeasures.jl, an easily extendable and highly performant open-source software that implements a vast selection of complexity measures. The software provides 1638 measures with 3,841 lines of source code, averaging only 2.3 lines of code per exported quantity (version 3.7). This is made possible by its mathematically rigorous composable design. In this paper we discuss the software design and demonstrate how it can accelerate complexity-related research in the future. We carefully compare it with alternative software and conclude that ComplexityMeasures.jl outclasses the alternatives in several objective aspects of comparison, such as computational performance, overall amount of measures, reliability, and extendability. ComplexityMeasures.jl is also a component of the DynamicalSystems.jl library for nonlinear dynamics and nonlinear timeseries analysis and follows open source development practices for creating a sustainable community of developers and contributors.

## 1 Introduction

A large aspect of nonlinear timeseries analysis [1–3] is concerned with extracting quantities (i.e. computing various statistics) from timeseries that quantify some property of the underlying dynamics that generated the timeseries. The purpose of these statistics can be to distinguish one type of dynamics from another [4], to classify timeseries into classes with different dynamics [5,6], to quantify directional associations between time series [7], which in turn can be integrated into frameworks for conditional independence testing between time series [8], and more. Most of these statistics are labeled *complexity measures*, because they quantify in some way the amount of complexity in the system. Although the word "complex" does

The codebase that produced them can be found in https://github.com/Datseris/ComplexityMeasuresPaper. Data collection and analysis method complied with the terms and conditions of the source of data.

**Funding:** UKRI's Engineering and Physical Sciences Research Council, grant no. EP/Y01653X/1. The funders had no role in study design, data collection and analysis, decision to publish, or preparation of the manuscript.

**Competing interests:** The authors have declared that no competing interests exist.

not have a widely-accepted definition yet [9], it typically describes something that is both not regular nor purely stochastic. There is a multitude (100+) of software that implement some form of complexity measures. An excellent summary of some of these software is given in the supplementary material of [10].

The majority of complexity measures are based on some form of axiomatically well-founded entropy. For example, the permutation entropy [11] and the wavelet entropy [12] are based on the Shannon entropy (Eq 1). Other complexity measures are not entropies in the formal mathematical sense, but are inspired by, or related to, entropies. Approximate entropy [13] and sample entropy [14], for example, are entropy *rates*, rather than entropies. Like an entropy, these entropy-like measures will typically yield higher numerical values for more "complex" data, where "complex" has a measure-specific definition. In the rest of the text we will refer to all these entropy or complexity quantities simply as *complexity measures*, and only use the word "entropy" if its rigorous mathematical definition is important in the context.

## 1.1 Example of estimating a complexity measure

There exists a surprisingly large amount of complexity measures in the literature. For simplicity, we will start by focusing on the largest class of complexity measures: those that are functionals of probabilities mass functions (PMFs). Here, we will deal exclusively with *empirical PMFs*, which are PMFs estimated from data. We will use the terms "probabilities" and empirical PMFs interchangeably.

All discrete probability-based complexity measures apply the same fundamental steps, and can be unified under one estimation pipeline. To estimate probabilities from observed data, it is necessary to first define a specific and countable *outcome space* $\Omega$. The goal is then to assign a probability $p_i = p(\omega_i)$ to each outcome $\omega_i \in \Omega$ based on the input data. To do so, the input data needs to be mapped (encoded, or discretized) onto the elements of $\Omega$. Sometimes, this procedure is also called *symbolization*. After encoding, we can form an empirical distribution over the encoded symbols. An empirical distribution in this context just means a histogram, or a normalized pseudo-histogram, depending on the structure of the outcome space. Next, the probabilities $p(\omega_i)$ are estimated from this empirical distribution, for example using relative frequency estimation. Finally, once a probability vector $\mathbf{p} = \{p(\omega_i)\}_{i=1}^N$ has been constructed, these probabilities can be given as input to some probabilities functional.

As an example, let's say we want to compute the order-3 permutation entropy [11] for an input time series $x$. To do so, we first construct a 3-dimensional embedding of $x$. Three-element state vectors can be ordered in 3! = 6 different ways. We'll consider each one of these possible orderings, which are also called *ordinal patterns*, as separate outcomes. We can then proceed by mapping each state vector in the embedding onto one of the ordinal patterns which is the same as saying that we *encode* the input data. To estimate probabilities, we can then simply count the relative frequency/occurrence of the different ordinal patterns and normalize these counts to sum to 1 (also called plug-in, or maximum likelihood estimation). Finally, these probabilities are given to the Shannon entropy formula [15]

$$H_S(p) = -\sum_i^M p_i \log(p_i). \tag{1}$$

This *estimator* of the Shannon entropy is called the *naive*, or *plug-in* estimator, and returns a nonnegative number which is indicative of the "complexity" of the input data.

## 1.2 Combinatorial explosion of complexity measures

In the first step of the procedure outlined above, we implicitly chose an *outcome space* (a way to map data into outcomes). We could use any other outcome space instead. One commonly used class of outcome spaces are rectangular binnings (i.e., histograms), in which each data point is mapped onto one bin according to its value. Other examples of outcome spaces are dispersion patterns [16], binned cosine similarities [17], binned state vector distances [18], or sorting complexity [19], which all consist of an initial embedding step, after which the embedding vectors are encoded using some procedure that cleverly highlights some interesting property of the underlying data.

In the second step, any count-based probabilities estimator could be applied to transform the observed outcome frequencies into probabilities. More sophisticated estimators include Bayesian regularization [20], shrinkage estimators [21], and add-constant estimators [22], which apply smoothing to the counts.

In the third step, we could have instead considered any of the other theoretically well-founded information measures, for example Rényi entropy ($H_R$) [23], Tsallis entropy ($H_T$) [24], Kaniadakis entropy ($H_K$) [25], Curado entropy ($H_C$) [26] and the Anteneodo-Plastino stretched exponential entropy ($H_{AP}$) [27], the lesser known Shannon extropy ($J_S$) [28], Rényi extropy [29] or Tsallis extropy ($J_T$) [30], fluctuation complexity [31], or any other probabilities functional that in some way quantify complexity.

Since the naive plug-in estimator systematically underestimates the Shannon entropy [32], in the third step, we could also have used any of the plethora of bias-corrected estimators for Shannon entropy that have been proposed [20,32–38]. Any of the other measures can also be computed either using plug-in estimation or other generic estimators such as the jackknife estimator [39], or any tailored measure-specific estimator.

Thus, excluding parameterizations, there are four degrees of freedom when computing a discrete, probabilities-based complexity measure: the discretization/encoding procedure, the probabilities estimator, the information measure, and the *estimator* for the information measure. Now let's assume that the literature describes $N_O$ different outcome spaces, $N_P$ different ways of estimating probabilities, and $N_C$ different probabilities-based complexity measures. Assume that the number of estimators for a particular measure $M_i$ is $N_{M_i}$. Then the number of total computable PMF-based complexity measures is

$$N_O \times N_P \times \sum_{i=1}^{N_C} N_{M_i} \qquad (2)$$

We quickly realize that there is a vast set possible complexity measures, varying across all four degrees of freedom, with differences ranging from minor technicalities to major conceptual ones, yet all quantifying complexity in some unique way. Adding (for example) one more entropy definition $N_C \rightarrow N_C + 1$ drastically increases the total number of computable measures, because there are $N_O \times N_P$ potential ways of computing it from data for every unique estimator of this new measure. We call this the *combinatorial explosion* of complexity measures. In this context, the traditional software design approach of implementing one function for each measure is *not scalable* as it requires adding many more functions when wanting to add "only" one more complexity measure definition.

The large number of possible complexity measures could provide great opportunities for future research into complexity quantification. However, a systematic and easy-to-use software for exploring and comparing these different measures, which is also easy to extend with new ones and has the computational capacity to compute all of them quickly, is lacking.

### 1.3 Enter ComplexityMeasures.jl

ComplexityMeasures.jl is recently developed Julia-based [40] software that fills this gap. We chose the Julia language because it is performant, high level, has an overwhelmingly open source community, and has a mature and feature-full software ecosystem that supports non-linear timeseries analysis (§2.5). In several scientific domains authors provide similar arguments for the adoption of Julia for computational research [41–43].

ComplexityMeasures.jl resolves the explosion problem by taking a fundamentally different approach: the software allows for *composable instructions* for how to compute a complexity measure. For discrete, probabilities-based measures, this entails composing instructions on which definition to use, which estimator to use, with which probabilities estimator to estimate probabilities, and which outcome space to use for the discretization (Figure 1). A similar approach is taken when estimating other complexity measures, which are described later in the paper. In practice, this leads to an incredibly lean source code base that is also easily extendable. This new design approach avoids the one-function-per-estimator typical software design that leads to an unnecessarily large, hard-to-maintain code base that is also hard to bug fix efficiently or future-proof. By developing ComplexityMeasures.jl following highest standards in scientific software development [44], we not only made the software capable of handling all present and future complexity measures, but also made it highly performant.

In Sect 2 we expose the design behind ComplexityMeasures.jl and how it integrates into a wider ecosystem enabling further innovation (such as Associations.jl [45]); in Sect 3 we provide some selected examples that highlight how accessible yet powerful the library is; in Sect 4 we compare the software with existing alternatives in a comprehensive tabular format and show that across many objective aspects of comparison ComplexityMeasures.jl performs best;

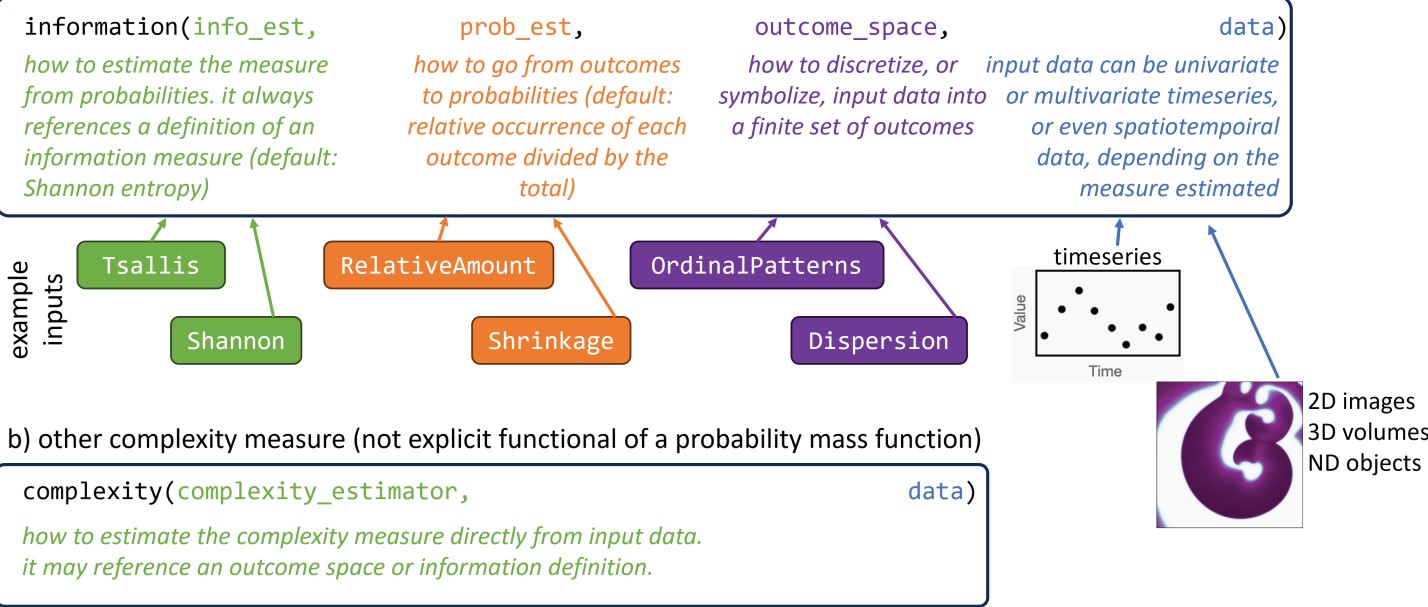

**Fig 1. The two central functions of ComplexityMeasures.jl and the abstract inputs they expect, along with a couple of concrete example inputs.** The full list of concrete possible inputs is displayed in Table 2. A runnable concrete code snippet that performs real-world analysis using ComplexityMeasures.jl can be seen in Appendix 5.

and lastly in Sect 5 we first summarize our work and present a list of the novelties and unique advantages of ComplexityMeasures.jl, and conclude with an outlook on the role of open source in the wider research community, our approach on promoting community efforts, and an invite to authors developing new complexity measures.

## 2 Design of ComplexityMeasures.jl

### 2.1 Core functions

The design of ComplexityMeasures.jl is displayed in Fig 1 and perfectly parallelizes the mathematically rigorous formulation of an information measure as we described it in Sect 1.1. Software usage revolves around two functions: `information` and `complexity`. The first is called as `information(discr_measure_est, prob_est, ospace, data)`, and estimates an information measure given the information measure estimator from a PMF, a probabilities estimator to map discrete outcomes into a PMF, and an outcome space to map input data into discrete outcomes. Internally in the software, this system is based on a functional approach that utilizes the multiple dispatch paradigm of the Julia language. When `information` is called many lower-level functions are called in sequence. The first function transforms the `data` to a symbolic representation using `ospace`. Multiple dispatch decides which concrete function implementation is used based on the type of `ospace`. Next, the symbolic representation is transformed into a PMF based on the type of `prob_est`. Finally, the estimated PMF is used to calculate an information measure, e.g. an entropy, based on the type of `discr_measure_est`. All other functions of ComplexityMeasures.jl operate in the same functional principle that utilizes multiple dispatch.

For an example of usage, to estimate the permutation entropy (as originally defined in [11]) one would call `information(Shannon, RelativeAmount, OrdinalPatterns(m = 3), input)`. For convenience and conciseness, simpler "shortcuts" are also possible in ComplexityMeasures.jl. For example, `entropy(OrdinalPatterns(m = 3), data)` uses default values for the probability estimator (`RelativeAmount()`) and the information measure estimator (`Shannon()`). For a few handpicked measures even simpler syntax is available, such as the call `entropy_permutation(data; m = 3)`, which is equivalent with the previous ones. More syntax shortcuts are discussed in the software documentation [46].

For complexity measures that are not explicit functionals of a PMF, we have implemented a simpler design shown in Fig 1b. There, a function `complexity` takes as input a "complexity estimator" that defines a complexity measure and includes instructions on how to compute it directly from input data. Differential (or continuous) information measure estimators follow the same design in the current software version. In the future, we plan to make a dedicated probability *density* estimation interface for continuous measures, similar to our discrete estimation interface based on probability mass functions. Both `information` and `complexity` have a normalized option that returns the complexity measure divided by its maximum possible value. Similarly, a `multiscale` function computes the multiscale variant of a chosen complexity measure (a framework originally introduced in Ref. [47] as "multiscale sample entropy"). Both normalized and multiscale forms are valid for *any* complexity measure in the library since these forms are also based on a composable design.

Structuring ComplexityMeasures.jl on this interface comes with many benefits:

1. Orthogonality of inputs (information estimator, probabilities, and outcomes). This means that if, e.g., a combination of a probability estimator works for a given outcome space, it is guaranteed to work for *any* outcome space implemented in the software. This

guarantee occurs automatically due to the design, and is made possible by the Julia language multiple dispatch system [40]. It does not need to be enforced by the user or even the developer.

2. The software is easy to maintain. Finding a bug, for example in the process of creating a histogram of some (optionally symbolized) data, requires fixing this bug in only a single function that does the histogram counting, not in potentially hundreds of functions that utilize histograms in some way.

3. Simple and scalable extensions. Adding a new outcome space requires writing code for one new type (Julia's version of "classes") and one mandatory function extension instructing how to discretize data for this outcome space. This can be as simple as 10 lines of code. Yet, once implemented, this outcome space would allow the user, without writing any additional code, to compute any probabilities-based complexity measure using this outcome space, in combination with any probabilities estimation, any information measure, and any estimator of this measure, including even some non-information complexity measures such as missing patterns [48].

4. The probabilities themselves are directly accessible by the user and compose a comprehensive interface that we expand more in Sect 2.2. Analyzing the probabilities directly may expose something interesting about the data that is "integrated away" by the complexity measure computation. For example, if one computes the associated probabilities for the 6 order-3 ordinal patterns of a timeseries coming for a logistic map, 1 of the 6 patterns will always have 0 probability due to the logistic map's dynamics. Information like this can only be obtained by looking at the probability mass function directly, which is hidden away in the majority of alternative software. Additionally, having access to the probabilities directly allows expanding and creating new complexity measures not published before like the example we showcase in Sect 3.2.

## 2.2 Outcome spaces and probabilities

The majority of complexity measures are estimated based on some PMF extracted from data. To create the extendable design mentioned above, behind the main functions `information`, `complexity` there exists a fully-fledged API (application programming interface) for extracting probabilities from data, based on the mathematically rigorous formulation of an outcome space. No other software on complexity analysis provides such an extensive interface for extracting probabilities from data.

In ComplexityMeasures.jl we define an abstract hierarchy of types called `OutcomeSpace`. An instance `o` of a concrete implementation of an `OutcomeSpace` describes how to discretize data into discrete outcomes. Given `o`, we define many functions for handling probabilities. For example, `total_outcomes` returns the cardinality of `o`, while `missing_outcomes` returns the number of outcomes defined by `o` as possible but not present in the data, and `counts` returns a vector of integers, counting the number of times each outcome was present in the input data. We further separate this API into outcome spaces that are counting-based, which allow mapping each element of input data into an integer, and into non-counting-based outcome spaces, which cannot do this mapping. More details on this can be found in the developer documentation of ComplexityMeasures.jl [46].

## 2.3 Clarifying and educative approach to naming

The literature is full of complexity measures that are named similarly, but represent fundamentally different concepts. For example, the term "bubble entropy" [19] is not an entropy per se, but a scaled difference between two Rényi [49] entropies which have been computed

by discretizing the input data in a particular manner which has to do with the bubble sort algorithm. Rényi entropy, however, is an axiomatically well-founded entropy. Similarly, permutation entropy [11] or wavelet entropy [12] are not distinct forms of entropy in terms of the mathematical definition of entropy. They are just the Shannon entropy computed by discretizing the input data into outcomes based on permutation patterns or wavelet coefficients, respectively. In fact, it is common for papers that introduce new discretization procedures, i.e. *outcome spaces*, to name these as some sort of "entropy" [16,17,50].

We believe that this abuse of terminology can be confusing, especially for newcomers to the field. Indeed, while teaching nonlinear timeseries analysis, we experience that students often interpret the Shannon entropy and permutation entropy as two different quantities. With ComplexityMeasures.jl and this paper, we aim to clarify this naming confusion, while also highlighting what the common elements between different complexity measures are: estimating probabilities from data. That is why in the software we do not promote function names like "permutation entropy". That being said, we understand that some terms (like the permutation entropy) are very well recognized in the field and should be more accessible. Thus, for a few handpicked complexity measures we provide a shorter syntax, such as the function `entropy_permutation(input; m = 3)`. Nevertheless we make it clear in the documentation of these "convenience functions" that they do not provide a genuinely new entropy even if named as such.

## 2.4 Software quality

ComplexityMeasures.jl was implemented following best practices in scientific code [44]. The software is free and open source (MIT-licensed), hosted on GitHub, and also available through the Julia package manager. The repository is composed in total of 3,841 lines of source code, 2,425 lines of test code, and 6,020 lines of documentation text according to PackageAnalyzer.jl [51]. The online documentation is very extensive [46]. It features a central tutorial; full API reference listing outcome spaces, entropy/complexity measures, and estimators provided by the software; more than a dozen of individual examples that are created by showing real code tied with its output; a developer's documentation for contributing more features to the library. The documentation also cites all research articles introducing the implemented measures/estimators via BiBTeX, and provides an explanation and/or description of each measure/estimator implemented, making it straightforward to understand what this measure is and what it estimates without needing to consult the research article. The software is extensively tested, with coverage of ~90%, which means that at least 90% of the source code lines are explicitly called in the test suite. Both tests and documentation are run through continuous integration upon every committed change to the software. ComplexityMeasures.jl is continuously improved and follows agile development practices [52]. A new feature, or even the smallest bugfix, is immediately released as a new software version, which the users can obtain instantly via a standard update command provided by the Julia language. The quoted numbers in this subsection refer to version v3.7 of the software.

## 2.5 Part of a greater whole

ComplexityMeasures.jl can be used as a standalone software. However, it is also part of a greater whole. It is a component of the DynamicalSystems.jl [53] software library for nonlinear dynamics and nonlinear timeseries analysis, and is also the basis for the Associations.jl library for relational (or causal) timeseries analysis [45]. In this way, ComplexityMeasures.jl integrates with a wider ecosystem for timeseries and data analysis.

For example, it can be immediately used with TimeseriesSurrogates.jl [54] to perform surrogate analysis testing for nonlinearities, thus eliminates the need to re-implement surrogate testing (as in e.g., in PyBioS [55]). Indeed, if the reader has a look at the associated code of Fig 3 in Appendix A1, performing timeseries surrogates significance testing by combining TimeseriesSurrogates.jl and ComplexityMeasures.jl is as seamless as if they were one package. DynamicalSystems.jl also has a component for fractal dimensions [2,Ch. 5]. Fractal dimensions themselves are complexity measures, and were initially part of ComplexityMeasures.jl. They were split off due to their codebase becoming extensive as well as being dedicated to a review article on fractal dimensions [56] (hence, in Table 1 we include these fractal dimensions as part of the comparison). Finally, a component of DynamicalSystems.jl is about estimating optimal parameters for delay coordinate embeddings [2, Ch. 6], including the latest methods in the literature [57]. This can be used seamlessly to estimate optimal delay time and/or embedding dimension, which are crucial for the majority of complexity measures. Additionally, DelayEmbeddings.jl is used by ComplexityMeasures.jl to perform the actual delay embedding, which is a benefit since DelayEmbeddings.jl has been optimized for delay embeddings all the way down to machine code.

A software based on ComplexityMeasures.jl is Associations.jl [45], which implements measures for relational (cross-variable) association quantification. Relational association quantification is in itself is a huge research field, extending far beyond the scope of ComplexityMeasures.jl, where we exclusively deal with quantifying complexity within a single dataset, not between datasets. Many of these cross-variable (conditional) measures can be expressed in terms of single-variable complexity measures. For example, conditional mutual information (CMI) can be decomposed as a sum of four marginal entropy terms computed from some subset of the joint variables considered for the relational association analysis. Each entropy estimator compatible with multivariate input data is therefore also a CMI estimator. We have been very deliberate in the design of ComplexityMeasures.jl to mimic the mathematical identities between, for example, CMI and entropy. In the case of CMI, any probabilities estimator or entropy definition added to ComplexityMeasures.jl automatically enables a corresponding CMI estimator upstream in Associations.jl. This enables limitless extensibility with no additional coding effort. We leave a detailed review of Associations.jl and its tools for relational/association analysis for a future paper.

## 2.6 Performance optimizations

A large part of the development time of ComplexityMeasures.jl has been spent exclusively on optimizing the software performance. Much of the optimization stems from performant algorithmic choices that could be done in any programming language. Due to the sheer amount of algorithms included in ComplexityMeasures.jl, listing each one of this is unfeasible. Instead, we provide a few examples of such choices in the comparison section (Sect 4).

The rest of the performance optimizations we employ are what we believe to be standard recommendations for writing performant Julia code, and most of them are discussed in the Julia manual. In summary, we have employed the following strategies.

- *Ensured type stability in all function and type definitions*. Type stability allows the Julia compiler to generate highly optimized machine code, because it can determine the exact type of every variable at compile time, significantly reducing overhead and increasing performance.
- *Minimized memory allocations throughout the code base*. Memory allocations can be significant performance bottlenecks in scientific computing. We have attempted to

**Table 1. Comparison table across software for entropic and complexity timeseries analysis.** The symbols mean: ✓ = has aspect, ✗ = does not have aspect, ◗ = partially has aspect. The numeric superscripts in the first column correspond to more extensive descriptions that we provide in the main text of Sect 4. The codebase that produced the benchmarks can be found online in [66]. Benchmarks were run on a laptop with 11th Gen Intel(R) Core(TM) i7-1165G7 at 2.80 GHz.

| | ComplexityMeasures.jl v3.7 | EntropyHub v2 | CEPS v2 | PyBioS (no v) |
|---|---|---|---|---|
| Software and development aspects | | | | |
| Language | Julia (and Python, R, C, due to interoperability) | Julia, Python, MATLAB | MATLAB | Python |
| Cost-free | ✓ | ✓ | ✗ | ✓ |
| OSI license | MIT | Apache-2.0 | LGPLv3 | not open source |
| GUI Interface | ✗ | ✗ | ✓ | ✓ |
| Multivariate input | ✓ | ✓ | ✗ | ✗ |
| Spatiotemporal input | ✓ | ◗ | ✗ | ✗ |
| Integrates with a wider ecosystem | ✓ | ◗ | ✗ | ✗ |
| Tests[1] | 92% coverage | no tests | no tests | no tests |
| Extendable design | ✓ | ✗ | ✗ | ✗ |
| Compiled documentation | ✓ | ✓ | ✗ | ✗ |
| Introductory tutorial | ✓ | ✗ | ✓ | ✗ |
| Explanation of measures in the documentation | ✓ | ✗ | ✓ | ✗ |
| Number of *executed* code examples in docs[2] | 19 | 13 | 0 | 0 |
| Developer's docs | ✓ | ✗ | ✗ | ✗ |
| Overall content (related to complexity measures, ignoring statistical, physiological, or relational measures) | | | | |
| Total entropy and complexity measures[3] | 1,638 (Appendix A2) | 26 | 74 | 7 |
| Outcome spaces | 13 | 0 | 0 | 0 |
| Probabilities estimators | 4 | 1 | 1 | 1 |
| Information definitions[4] | 12 | 0 | 0 | 0 |
| Entropy estimators | 7 | 1 | 1 | 1 |
| Fractal dimensions | 11 | 0 | 17 | 0 |
| Normalized forms | ✓ | ◗ | ✗ | ✗ |
| Multiscale functions | 2 | 4 | 1 | 0 |
| Performance comparison[5] | | | | |
| Permutation entropy | 0.56 ms, 470.47 KiB | 7.16 ms, 8.89 MiB | - | - |
| Sample entropy | 0.5 s, 1.71 MiB | 5.23 s, 7.63 GiB | - | - |
| Cosine similarity entropy | 1.74 ms, 2.04 MiB | 5.03 s, 4.48 GiB | - | - |
| Dispersion entropy | 1.46 ms, 525.75 KiB | 78.55 ms, 8.23 MiB | - | - |
| 30-D value histogram | 1.47ms, 177.70 KiB | ∞ (out of memory) | - | - |

eliminate unnecessary allocations by pre-allocating storage containers (e.g. `Array`s) and reusing existing memory structures where possible. As a result, memory allocations were reduced by orders of magnitude compared to competing implementations (Table 1). Reduction in memory allocations directly translates to faster execution times and better scalability for larger datasets.

- *Separated performance-critical code into dedicated functions.* This technique, known as function barriers, allows the compiler to optimize each critical section independently and make better inlining decisions.

- *Utilized StaticArrays.jl for fixed-size vector operations*. Static arrays are stack-allocated rather than heap-allocated, providing immediate performance benefits through better cache utilization and zero garbage collection overhead. Additionally, their fixed size allows the compiler to perform aggressive optimizations including SIMD vectorization and loop unrolling.
- *Delegated performance-critical algorithmic steps to highly optimized and specialized Julia packages*. For example, we use Distances.jl for metric space computations and DelayEmbeddings.jl for delay embeddings.

Finally, we purposely did not parallelize anything in ComplexityMeasures.jl. It is simply more efficient, and simpler from a source code perspective, to parallelize at the high level (such as when looping over different parameters or input timeseries or surrogate timeseries). Indeed, TimeseriesSurrogates.jl, a companion software discussed in §2.5, is parallelized via multithreading at a higher level.

As becomes evident in the comparison with other software (Sect 4), these performance optimizations and algorithmic choices, along with generally following good development practices for Julia code, make ComplexityMeasures.jl by far the most performant software for complexity measure estimation, sometimes >1,000x faster than the competition (Table 1).

## 3 Example applications

### 3.1 Simple complexity analysis of stock market timeseries

Here, we present a straightforward analysis of stock market timeseries from the lens of complexity measures. This is an emerging topic in the literature, with relevant publications emerging only in the last 5 years. For example, Refs. [58,59] show that regularity of the stock market anti-correlates with sample entropy. This example highlights how simple it is to integrate ComplexityMeasures.jl within a realistic data analysis workflow, as well as how few lines of code the user needs to write: at most 1 line of code per complexity measure estimated.

For this analysis, we used a range of complexity measures: sample entropy [14], approximate entropy [13], permutation entropy [11], dispersion entropy [16], and spectral entropy [50]. These measures were estimated for the 500 stocks that compose the S&P500 index, using daily-resolution timeseries of stock closing prices for the years 2000 to 2020. The complexity measures were then compared with the relative success of each stock with respect to S&P500, that is, the total price change of the stock divided by the total price change of S&P500 from start to end of the time interval (Fig 2). The analysis shows that sample, approximate, and dispersion entropy anti-correlate with stock success, spectral entropy correlates with stock success, and permutation entropy has no relation with stock success. The code for this analysis is also in Appendix 5.

This example is a modification of a group project performed by BSc Mathematics students over the course of 4 weeks. The students did not have any prior knowledge of the Julia programming language, nor of the concepts of complexity measures. This is a testament to how simple it is to learn and use ComplexityMeasures.jl, even if one has to learn an entirely new programming language from scratch.

### 3.2 Missing patterns research acceleration

In Ref. [48] the authors devise a complexity measure that can be used in timeseries surrogate studies [54,60] to detect nonlinearity in a timeseries. This measure is called *missing dispersion patterns*, and is estimated as follows. First, a timeseries is mapped onto outcomes defined

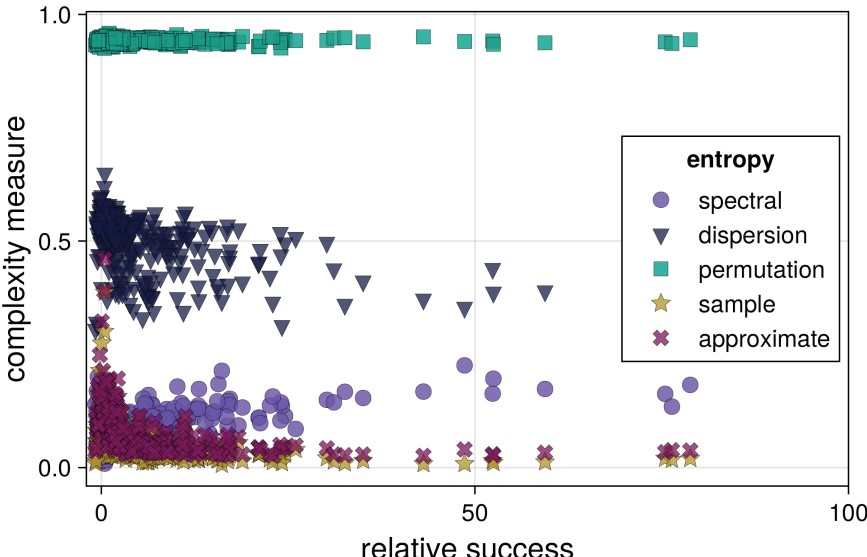

**Fig 2. Stock market timeseries analysis: various complexity measures of each stock plotted versus the relative success of the stock with respect to S&P500 index.**

by the *dispersion patterns* outcome space. This outcome space was used by Ref. [16] to define the *dispersion entropy*. We then count how many of the total possible outcomes (dispersion patterns) are actually present in the data. The ones not present are the missing dispersion patterns.

The concept of missing dispersion patterns is trivially generalizable to any outcome space. Instead of missing dispersion patterns one has missing outcomes. Indeed, besides the missing dispersion pattern, Ref. [48] estimated also the missing ordinal patterns (used to define the permutation entropy [11]) and compared the two.

In this subsection we show how easy it is to make such generalizations, and accelerate novel research with ComplexityMeasures.jl. The software implements the generic function `missing_outcomes`, which can take in any outcome space to obtain the missing outcomes. With this function, one can create a complexity measure similar to the missing dispersion patterns of Ref. [48], but for *any* outcome space. Perhaps other outcome spaces, not explored in Ref. [48], are more suitable for distinguishing nonlinearity than the dispersion patterns, and such new research would be very easy to do with ComplexityMeasures.jl. We show an example of such an analysis in Fig 3, and the code to create the figure in Appendix A1. The results show that missing dispersion patterns can detect nonlinearity in simple nonlinear systems such as the logistic map, and the same can be said for missing ordinal patterns. However, neither of the two appear capable of detecting non-linearity in a moderately complex 8-dimensional chaotic system (the Lorenz-1996 model [61]). These results can lead to new research with more careful analysis that gives a more robust estimate of whether missing outcomes can be used to distinguish nonlinearity.

### 3.3 Paintings in the complexity entropy plane

The complexity-entropy plane is a way to characterize data in terms of their complexity [62]. Ref. [63] analyzed the history of art paintings by quantifying in terms of this complexity plane. They consider a database of ca. 137,000 historical artworks from the `WikiArt.org`

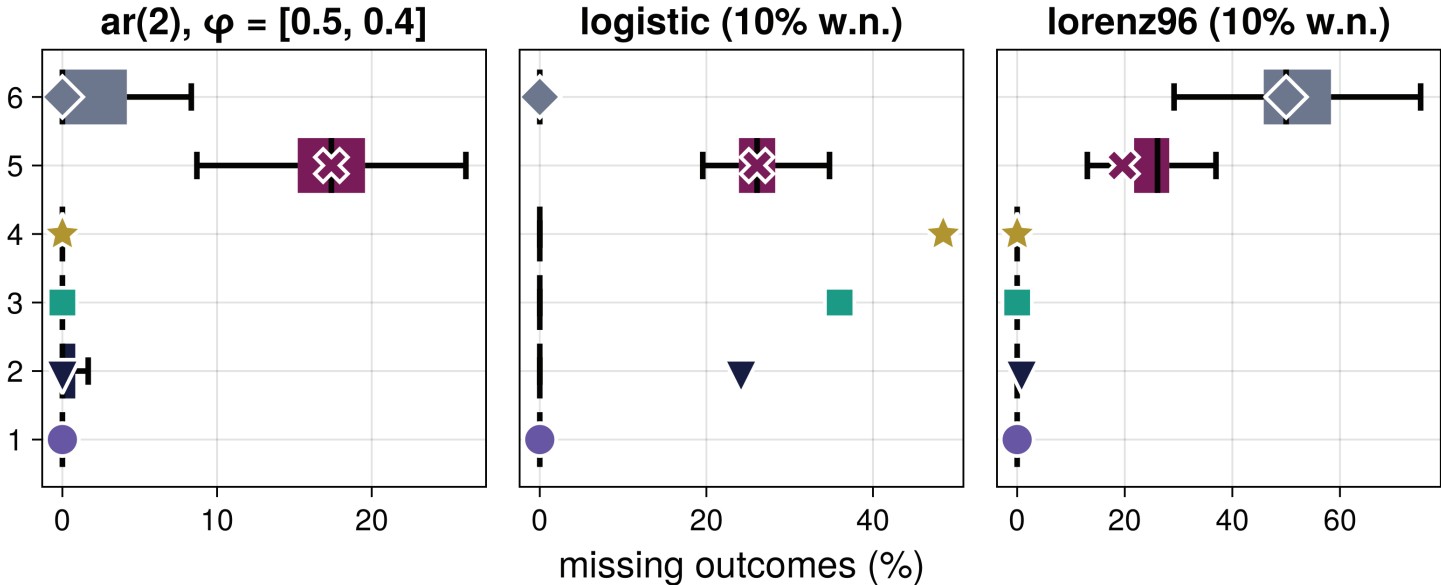

outcome space: 1 = ordinal (m = 4), 2 = ordinal (m = 5), 3 = dispersion (m = 2, c = 5),
4 = dispersion (m = 3, c = 4), 5 = bubble sort swaps (m = 10), 6 = cosine similarity (m = 3, nbins = 24)

**Fig 3. Using missing outcomes to test for nonlinearity.** The figure shows three panels for three timeseries: a stochastic autoregressive process of order 2, the chaotic logistic map with 10% white noise, and an 8-dimensional chaotic Lorenz-1996 model [61] with 10% white noise. For each panel we estimate the missing outcomes percentages (x-axis) corresponding to six outcome spaces (y-axis), with spaces numbered. 3 and 4 relating to Ref. [48]. The box plots show the percentage of missing outcomes of random surrogate realizations of the timeseries versus the value corresponding to the real timeseries (markers with white outlining). See [2, ch.7] for more about timeseries surrogates.

database, and convert each painting into a $n_x$-by-$n_y$ matrix $M$, where $n_x$ and $n_y$ are the pixel dimensions of the image, and the matrix entry $M_{ij}$ is a number between 0 and 1 obtained from a greyscale-like transformation of the red-blue-green color channels in the $ij$-th pixel. For each painting, they then compute normalized spatial (Shannon) permutation entropy [11] ($H_S^{perm}$) and statistical complexity [4] ($C$) from PMFs constructed by sliding a 2x2 square pixel window across the painting and counting the relative frequency of ordinal patterns.

ComplexityMeasures.jl provides an easily extendable interface for spatiotemporal probabilities and generalized entropies. With our implementation of the generalized statistical complexity measure [4,64], we can not only reproduce [63]'s analysis, but easily perform a much more varied study of artistic styles in terms of their complexity. Our implementation of the StatisticalComplexity measure is a prime example of the power of multiple dispatch of the Julia language: the estimator leverages the entire discrete entropy-based ecosystem. The measure accepts any spatial probabilities estimator, which can be arbitrarily parameterized (currently, we offer three such estimators), and accepts any stencil/template (local pixel arrangement) to construct the probability distributions over an image (or higher-dimensional arrays), not restricted to adjacent 2x2 or 3x3 pixel patterns. StatisticalComplexity also works with any normalizable discrete information measure definition and estimator, and accepts many different distance measures for computing $C$. This powerful interface allows us to explore the robustness of [63] results with relatively few lines of code.

Reproducing [63]'s ordinal pattern based analysis a 2x2 square stencil (Fig 4, left panel), supplementing with a custom non-square 4-pixel stencil using a dispersion patterns outcome

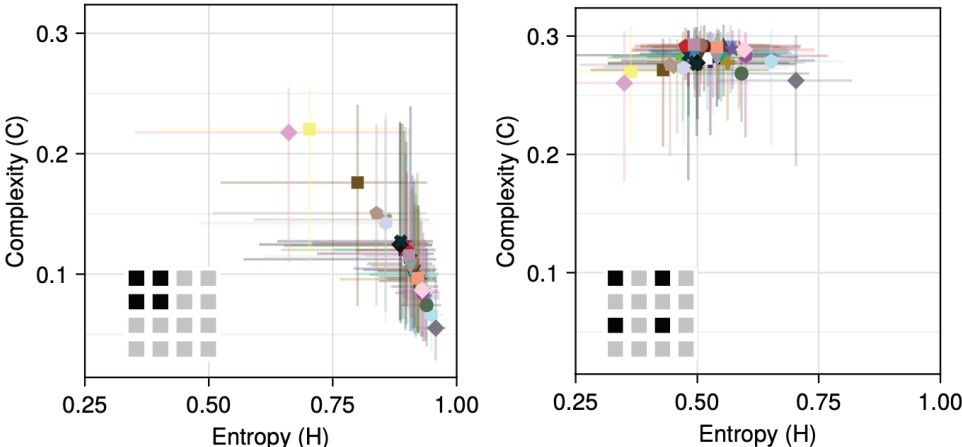

**Fig 4. Re-analysis of Fig 2 in [63], using our updated dataset of 153465 paintings (downloaded July 2023).** Only styles with at least 1000 paintings in the Wikiart database are included here. In the left panel, we've used the `SpatialOrdinalPatterns` (as in [63]), while in the right panel, we've used the `SpatialDispersion` outcome space with parameter `c = 3`. Insets show the shape of the pixel stencil. Note that [63]'s error bars are standard errors of the mean for each style, while here scatter points here are the medians with 10th to 90th percentile ranges.

space (Fig 4, right panel), we find that artistic styles *differ dramatically* in their *H-C* values depending on stencil pattern and outcome space, even when there are only small differences in the stencil pattern. Since [63]'s paper is not accompanied by code, we cannot actually determine whether their *H-C* values are exactly reproduced here, or if any differences are caused by the addition of more paintings to the database since their analysis, differing code implementations, or other factors. However, using our software, it is trivial to further explore these topics further using different outcome space, probabilities estimators, and entropy definitions/estimators, which we leave for future work.

## 4 Comparison with alternative software

There is a multitude (100+) of software that implement some form of complexity measures. An excellent summary of some of these software is given in the supplementary material of [10]. However, the overwhelming majority of these software only provide a dozen or so complexity measures as individual functions, and/or focus on a particular type of timeseries (e.g. physiological, EEG, or ECG timeseries). While we may have missed something, after a brief overview of these software, none of them appear to provide a composable orthogonal design like ComplexityMeasures.jl.

Here we compare the performance of ComplexityMeasures.jl with notable other softwares that not only offer complexity quantification, but also have an associated peer-reviewed publication and appear to have a decent number of features: EntropyHub [65], CEPS (complexity and entropy in physiological timeseries) [6], and PyBioS [55].

In Table 1 we showcase an extensive comparison between ComplexityMeasures.jl and these three software. Explanations to the superscripts in the table are as follows:

1. Surprisingly, even though all other software have been published through peer review in reputable journals, none of them has any *publicly accessible* test suite that confirms the correctness of the software. ComplexityMeasures.jl has an extensive publicly accessible test suite covering $\sim 90\%$ of the total source code of the software. Without (public) tests,

the only way to check for software validity is either for every user to create their own test suite, or to "just trust the developers". Either option dramatically reduces software reliability.

2. Executed code examples are examples in the documentation that are the result of running real code snippets during the compilation of the documentation, and presenting the executed code and its output interlaced in the documentation. They are the only way to absolutely guarantee that the syntax presented in the documentation, and the output it produces, are actually the result of running the software.

3. Counting the total number of measures in each package is non-trivial and may have different answers depending on how one counts. For the other software we counted the individual entries in the table of contents in their respective manuals, trying to count fundamentally different variants (e.g., amplitude-aware vs. standard permutation entropy) as different versions. We excluded cross-entropies from this list, since we implement cross (relational) measures in the Associations.jl software instead of ComplexityMeasures.jl. A rough estimate shows the cross-measures in Associations.jl to be in the several hundreds, compared to $\sim$dozen in the alternatives.

4. This row refers to the definition of "true" or axiomatic entropies. That is, permutation entropy does not count as a new entropy definition. ComplexityMeasures.jl is unique in this approach, as it allows the concept of entropy definition. CEPS, while it allows computing the Tsallis entropy and the Tsallis permutation entropy, it doesn't allow composing the Tsallis entropy definition with arbitrary probability mass functions.

5. For the performance comparison, we evaluated the permutation, sample, cosine similarity, and dispersion entropy of a white noise timeseries with embedding dimension $m = 4$, and the Shannon entropy of the histogram of a 30-dimensional chaotic Lorenz-1996 (Ref. [61]) timeseries. We report the computation time and the allocated memory, showing that ComplexityMeasures.jl is routinely 10-1000x faster. For CEPS and PyBioS we cannot measure performance straightforwardly, because they are GUI-based. We note that even a difference of 10x in performance can already have a massive impact in real-world applications; typically one wants to estimate a complexity measure for many input timeseries (like in Sect 3.3), or for thousands of surrogates of an input timeseries (like in Sect 3.2).

## 4.1 Comments on performance differences

We provide here some brief comments illustrating likely causes for the large performance differences between our software and the closest alternative written in the same programming language: EntropyHub. Some of the performance improvements in ComplexityMeasures.jl come from following good practices in writing performant Julia code that we discussed in Sect 2.6. For example, for the cosine similarity entropy, compared to EntropyHub, we achieve a 2891x speedup, resulting primarily from 2196x less memory allocations. These occur largely due to the usage of the optimized Julia package DelayEmbeddings.jl for the delay embedding step. It uses optimized code generation based on statically-sized stack-allocated arrays so that the whole delay embedding operation is unrolled by the compiler. Our sample entropy implementation has 4462x less memory allocations than for EntropyHub's implementation, resulting in a 10x faster runtime. This comes likely due to better memory management and re-using existing arrays. In addition we are using KD-trees for local neighbors searches rather than using naive brute-force distance searches.

Additional improvements in performance come from the use of different and more efficient algorithms to estimate the same quantity. For permutation entropy, we get significant

speed-ups due to efficient encoding of permutation patterns into integers, which are more efficient to use for computations, using the "Lehmer code" [67]. In general, integer encodings result in faster computations throughout the package, not only because operations on integers are simpler, but because we drastically reduce memory allocations for large input datasets compared to operating directly on arbitrary-dimensional state vectors. Histogram-based methods are so fast in ComplexityMeasures.jl not only due to standard Julia code optimization, but also because we use a new algorithm for estimating the histogram of high-dimensional datasets which does not store bins. The memory requirement of this algorithm does not scale exponentially with the dataset dimension as is typically the case. This algorithm is used when any sort of histogram of a delay-embedded timeseries or symbol sequence is required, which is in fact the majority of methods in the library. For example, our dispersion entropy algorithm is 53x faster than EntropyHub's implementation because it uses this histogram optimization algorithm. The algorithm is described in detail in Appendix A, Sect 1 of [56].

These discussions illustrate how a careful approach to the software performance, in terms of following best practices, choosing the best algorithms, or even inventing new ones, can add up to overall large speed ups.

## 5 Conclusions

In this paper, we introduced ComplexityMeasures.jl, the reasoning behind its design, and its efficacy with respect to alternatives. The advantages and unique features of ComplexityMeasures.jl can be summarized as follows:

- Modular and composable design based on the mathematically rigorous formulation of a complexity measure. This design allows computing complexity measures that have not even been published yet in the literature.
- Easily scalable in terms of implementing new measures devised in new research without writing much new code.
- Intelligent utilization of Julia's multiple dispatch system [40] to create an incredibly lean codebase averaging only 2.3 lines of code per potential measure to be estimated.
- Exceptional computational performance, up to 1,000x faster than alternatives. This enables practical applications that were previously limited by computational capacity, such as extensive null hypothesis testing using surrogate data on large timeseries ensembles.
- Extensively tested through automated testing (continuous integration), providing a trustworthy environment.

Due to these features, we believe that a wider adoption of ComplexityMeasures.jl can be an invaluable asset for both academics and industry, wherever estimating complexity measures is useful.

We also wish to highlight the openness in the development of ComplexityMeasures.jl. ComplexityMeasures.jl was purposefully built from the ground up on Github as a *community effort*. After we (the authors) crossed paths in various open source projects trying to implement similar functionality, we quickly realized that there were *many* commonalities between our disparate research fields. We also realized that the most efficient way forward to ensure solid, reproducible research, and to achieve a good overview of the field, was to establish a common software framework for complexity quantification. Since its conception more than four years ago, ComplexityMeasures.jl's Github repository has seen over 200 pull requests and

over 140 issues (public discussions) have been addressed, implemented, and closed. We follow best open-source practices and greatly value openness and community contributions (see Acknowledgments).

It is our hope that, in the future, researchers that develop new complexity measures contribute them directly to ComplexityMeasures.jl when submitting their paper for review. This has numerous benefits for both the researchers themselves, and for the wider community. For the researchers, their new method is instantly accessible to an established pool of users, and citable via BiBTeX integration, generating the contributing researchers citations and recognition. The method is also instantly integrated with a wide range of estimators, allowing further research and study. For the community, the method becomes part of a well-tested, well-documented, and established software, enhancing reliability and accessibility. The integration of ComplexityMeasures.jl with a wider ecosystem such as DynamicalSystems.jl also allows the community to easily test whether the claims of a new paper are accurate and robust w.r.t. variability in the parameters or input data (e.g., see the applications of Sects 3.2 and 3.3). Lastly, this approach also promotes openness in academic code, an aspect that we believe is important in its own right, beyond complexity measures.

## Appendices

**A1 Code snippet for Figure 3.** See the reproducible code repository [66] for code for all figures (and also code highlighting via GitHub).

```
using ComplexityMeasures # at least version 3.7
using TimeseriesSurrogates # at least version 2.7
using ARFIMA
using CairoMakie
using PredefinedDynamicalSystems
using Statistics using
Random

# %% Setup: decide which outcome spaces to use for missing outcomes:
ospaces = [ # map delay time to concrete outcome space instance
    tau -> OrdinalPatterns(; tau, m = 4),
    tau -> OrdinalPatterns(; tau, m = 5),
    tau -> Dispersion(; tau, m = 2, c = 5),
    tau -> Dispersion(; tau, m = 3, c = 4),
    tau -> BubbleSortSwaps(; tau, m = 10),
    tau -> CosineSimilarityBinning(; tau, m = 3, nbins = 24),
]

# %% Generate timeseries
N = 2000 # length of timeseries
rng = Xoshiro(124314) # reproducibility

# Logistic map timeseries
ds = PredefinedDynamicalSystems.logistic(r = 4.0)
Y, t = trajectory(ds, N-1; Ttr = 100)
y = standardize(Y[:, 1]) .+ 0.1 .* randn(rng, N)

# Lorenz96 timeseries
ds = PredefinedDynamicalSystems.lorenz96(8; F = 24.0)
Dt = 0.01
W, t = trajectory(ds, (N-1)*Dt; Dt, Ttr = 100)
w = standardize(W[:, 1]) .+ 0.1 .* randn(rng, N)

# Arma timeseries
phi = SVector(0.5, 0.4)
x = arma(rng, N, 1.0, phi)

# %% Main computation
# function that computes normalized % of missing outcomes
nmo(o, x) = 100missing_outcomes(o, x)/total_outcomes(o)
```

```
surrotype = AAFT() # amplitude-adjusted fourier transform

# Set up figure and axes
fig = Figure()
axs = [Axis(fig[1, i]) for i in 1:3]

# loop over timeseries and outcome spaces
for (i, t) in enumerate((x, y, w))
    # estimate delay time for embedding
    if i == 2 # logistic timeseries always has delay 1
        tau = 1
    else
        tau = estimate_delay(t, "mi_min")
    end

    # initialize a generator for surrogates
    sgen = surrogenerator(t, surrotype)
    for (j, ogen) in enumerate(ospaces)
        o = ogen(tau) # `o` is the concrete outcome space instance

        # Surrogate test allows us to compute the quantity of interest
        # (here missing outcomes) for input data and 1000 surrogates
        # using parallel computing
        stest = SurrogateTest(x -> nmo(o, x), t, surrotype; rng, n = 1000)
        # extract real value and surrogate values
        rval, vals = fill_surrogate_test!(stest)
        # plot the results:
        boxplot!(axs[i], fill(j, 1000), vals; orientation = :horizontal)
        scatter!(axs[i], rval, j; strokecolor = :white)
    end
end

display(fig)
```

**A2 Total measures in Complexity Measures**   In Table 2 we list the content of ComplexityMeasures.jl in terms of available outcome spaces, information and complexity measures, and estimators thereof. In this subsection we use this table to count how many total complexity measures can be estimated with ComplexityMeasures.jl, version 3.7.0. We advise the reader to visit the software documentation for always-up-to-date measure counts estimates that are computed fully programmatically from the software source code [46].

We start with estimating the ways to extract a PMF from data. PMFs are used to estimate discrete information measures or complexity measures. Currently, the software implements 16 outcome spaces. Ten of these count-based, and for each count-based outcome space, PMFs can be estimated using four different probabilities estimators. The six remaining outcome spaces can generate PMFs in one way (using the `RelativeAmount` estimator). Therefore, there are currently $10 \cdot 4 + 6 = 46$ different ways of estimating a PMF from data.

There are 12 discrete information measure definitions in the software. Every discrete information measure can be estimated with either of the two generic estimators `JackKnife` or `PlugIn`. For Shannon entropy, five additional dedicated estimators are implemented. This means that in total we have $12 \cdot 2 + 5 = 29$ ways to estimate a discrete information measure from a PMF. Since all discrete information measures are functionals of PMFs, the number of ways to estimate a discrete information measure from data is $29 \cdot 46 = 1334$.

Next, the software also implements 12 differential information measure estimators that compute Shannon entropy. One of these estimators, `LeonenkoProzantoSavani`, can also estimate Rényi and Tsallis entropies. This gives a total of $12 + 2 = 14$ ways to estimate a differential information measure from data.

Finally, the software provides 7 complexity estimators that are not functionals of PMFs. One of them, `StatisticalComplexity`, must be handled separately, because in our

**Table 2. Central abstract types in ComplexityMeasures.jl and their concrete implementations for software version 3.7.**

| Abstract type | Concrete implementations |
|---|---|
| `OutcomeSpace` | **Uni- or multivariate timeseries**: `UniqueElements`, `ValueBinning`, `OrdinalPatterns`, `WeightedOrdinalPatterns`, `AmplitudeAwareOrdinalPatterns`, `Dispersion`, `CosineSimilarityBinning`, `BubbleSortSwaps`, `SequentialPairDistances`, `TransferOperator`, `NaiveKernel`, `WaveletOverlap`, `PowerSpectrum`; **Spatiotemporal timeseries**: `SpatialDispersion`, `SpatialOrdinalPatterns`, `SpatialBubbleSortSwaps` |
| `ProbabilitiesEstimator` | `RelativeAmount`, `Shrinkage`, `BayesianRegularization`, `AddConstant` |
| `InformationMeasure` | `Shannon`, `Renyi`, `Tsallis`, `Curado`, `Kaniadakis`, `StretchedExponential`, `ShannonExtropy`, `RenyiExtropy`, `TsallisExtropy`, `FluctuationComplexity` |
| `DiscreteInfoEstimator` | **Generic**: `PlugIn`, `Jackknife`; **Shannon-entropy specific**: `MillerMadow`, `HorvitzThompson`, `Schuermann`, `GeneralizedSchuermann`, `ChaoShen` |
| `DifferentialInfoEstimator` | **Shannon entropy**: `KozachenkoLeonenko`, `Kraskov`, `Goria`, `Gao`, `Zhu`, `ZhuSingh`, `Lord`, `AlizadehArghami`, `Correa`, `Vasicek`, `Ebrahimi`; **Shannon, Rényi or Tsallis entropy**: `LeonenkoProzantoSavani` |
| `ComplexityEstimator` | `ApproximateEntropy`, `SampleEntropy`, `LempelZiv76`, `MissingDispersionPatterns`, `StatisticalComplexity`, `ReverseDispersion`, `BubbleEntropy` |

software, we made the innovation to make the statistical complexity computable from any configuration of input outcome spaces, probabilities estimators, discrete information measure definition and estimator. To be a bit conservative with counting, we here only count the variability of the statistical complexity arising from varying outcome spaces and discrete information measure definitions, as these two would have the strongest impact on the statistic. This gives $16 \cdot 12 = 192$ variants of statistical complexity, and the total number of non-probability-based complexity measures is thus $192 + 6 = 198$.

We believe that it is fair to count some of the probabilities functions themselves as additional measures (see Sect 2.2), because they allow straightforwardly defining new, or expanding existing, complexity measures, as we show in Sect 3.2. In particular here we count two functions `probabilities`, `allprobabilities`, as all other probabilities-related functions can be created based on them in ComplexityMeasures.jl. These two functions can be combined with any way of estimating PMFs from input data, which means that we have $2 \cdot 46$ additional "probabilities measures" as well.

This makes the grand total of measures that one can estimate with ComplexityMeasures.jl equal to: $92+198+14+1334 = 1638$. This total is also estimated programmatically in the online documentation of ComplexityMeasures.jl (so see the online documentation for an up-to-date estimation [46]).

## Acknowledgments

We would like to acknowledge all volunteer contributors to ComplexityMeasures.jl which can be found on its GitHub page under the URL: https://github.com/JuliaDynamics/ComplexityMeasures.jl/graphs/contributors. KAH would like to thank the Earth System Evolution group at the University of Bergen, in particular Bjarte Hannisdal and David Diego, for creating a thriving environment where scientific software development becomes a natural part of the research process.

## Reproducibility

The figures and the performance numbers quoted in the comparison table are fully reproducible. The codebase that produced them can be found in [66]. Data collection and analysis method complied with the terms and conditions of the source of data.

## Author contributions

**Conceptualization:** George Datseris, Kristian Agoster Haaga.

**Data curation:** George Datseris, Kristian Agoster Haaga.

**Formal analysis:** George Datseris, Kristian Agoster Haaga.

**Funding acquisition:** George Datseris.

**Methodology:** George Datseris, Kristian Agoster Haaga.

**Software:** George Datseris, Kristian Agoster Haaga.

**Visualization:** George Datseris, Kristian Agoster Haaga.

**Writing – original draft:** George Datseris, Kristian Agoster Haaga.

**Writing – review & editing:** George Datseris, Kristian Agoster Haaga.

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
