## [Decision Letter · Decision Letter 0]

5 Nov 2024

PONE-D-24-22899ComplexityMeasures.jl: scalable software to unify and accelerate entropy and complexity timeseries analysisPLOS ONE

Dear Dr. Datseris,

Thank you for submitting your manuscript to PLOS ONE. After careful consideration, we feel that it has merit but does not fully meet PLOS ONE’s publication criteria as it currently stands. Therefore, we invite you to submit a revised version of the manuscript that addresses the points raised during the review process.

 Both Reviewers found value in the work, highlighting that it adds to what currently exists in literature. However, they also raised some concerns that merit attention and to be carefully addressed. In particular, one of them suggest to insert more details about some technical and computational aspects that can be useful for the readers and for those who work in the same field. I would encourage the authors to address these points in order to to improve the overall quality of the manuscript.

We look forward to receiving your revised manuscript.

Kind regards,

Alessandro Mengarelli

Academic Editor

PLOS ONE

2. In your Methods section, please include additional information about your dataset and ensure that you have included a statement specifying whether the collection and analysis method complied with the terms and conditions for the source of the data.

“UKRI's Engineering and Physical Sciences Research Council, grant no. EP/Y01653X/1.”

4. We notice that your supporting information is included in the manuscript file. Please remove them and upload them with the file type 'Supporting Information'. Please ensure that each Supporting Information file has a legend listed in the manuscript after the references list.

Reviewers' comments:

Reviewer's Responses to Questions

**Comments to the Author**

1. Is the manuscript technically sound, and do the data support the conclusions?

Reviewer #1: Yes

Reviewer #2: Yes

2. Has the statistical analysis been performed appropriately and rigorously? 

Reviewer #1: Yes

Reviewer #2: Yes

3. Have the authors made all data underlying the findings in their manuscript fully available?

Reviewer #1: Yes

Reviewer #2: Yes

4. Is the manuscript presented in an intelligible fashion and written in standard English?

Reviewer #1: Yes

Reviewer #2: Yes

5. Review Comments to the Author

Reviewer #1: Manuscript No. PONE-D-24-22899

ComplexityMeasures.jl: scalable software to unify and accelerate entropy and complexity timeseries analysis.

In this work the autors have introduce ComplexityMeasures.jl. It is an open-source software designed to efficiently implement a wide range of complexity and entropy measures for nonlinear time series analysis. They claimed that present software 1530 measures in just 3,834 lines of code. It indicates that, averages 2.5 lines per measure (version 3.5) due to its mathematically rigorous and composable design. This software offers superior computational performance, reliability, and extensibility compared to alternatives. As part of the DynamicalSystems.jl library, it supports sustainable development practices and provides researchers with a powerful tool for selecting appropriate measures and accelerating complexity-related research.

In conclusion, ComplexityMeasures.jl offers a scalable and high-performance solution for computing a wide range of complexity measures, outperforming alternatives by up to 1,000x and enabling practical applications previously limited by computational resources. Its composable design allows for easy creation of new measures, and its extensibility ensures instant usability across the broader ecosystem. Built as an open-source community project on GitHub, ComplexityMeasures.jl emphasizes openness, collaboration, and reproducibility, with over 200 pull requests and 140 closed issues since its inception. We encourage researchers to contribute new complexity measures directly to the software, benefiting both individual researchers by providing instant accessibility and recognition, and the wider community by enhancing reliability and testing within the DynamicalSystems.jl ecosystem. This approach not only improves the accuracy of new research but also promotes the openness of academic code, an important principle for scientific progress.

This should be immediately accept for publication.

Reviewer #2: The paper presents a novel software solution ComplexityMeasures.jl for complexity measures in timeseries analysis. The novel software includes a vast number of measures, while the authors claim high computational performance, reliability, and extendability. The paper is interesting and it clearly presents a fine addition to the body of knowledge.

The Introduction is rather broad, and nicely presents the motivation and the rationale for complexity measures calculation showing its relevance. It starts with a discussion on entropy and its relation to complexity measures, giving an example of complexity measure and focusing on the problem of computability due to combinatorial explosion.

However, I would like to see Section 1.3. a bit more expanded. It emphasizes novelty of the paper with the innovative software design. However, I think it should also emphasize 2-3 main contributions of the paper, bullet-by-bullet. Also, some review of the other existing solution would be welcome in the introductory part, as well, not only in Section 4, where it is more focused on comparison.

Fig 1. is not really a figure, but merely a description of the prototype of the function. I think iy can be transformed into some kind of a diagram that can better depict the two central functions of the implemented ComplexityMeasures.jl. I think the composable structure of the solution should be presented graphically. Even some classification of complexity measures by type or elements that are needed in implementation in a form of a table would help to understand some design decisions. There are some discusions on it on lines 132-139.

Section 2 is missing information on the framework used to implement ComplexityMeasures.jl. Although I can assume it is written in Julia, I think the reader should not base his toughts on this kind of "assumptions", but should be explicitely given relevant details. Specifically, you should also discuss why you used Julia, and any elements of parallel programming if you used them in implementation.

In section 2.2, you state "To the best of our knowledge, there isn’t any open source software in any programming language that provides such an extensive interface for extracting probabilities from data.". Can you back it up with some references to other APIs or frameworks, alternatives? Notably in Python and R?

In several places you point out to the documentation of ComplexityMeasures.jl. Provide a link, and put it in the list of references.

Section 2.4. Provide a link to github project and reference it here.

Section 2.6. is too short. I think that the authors should provide the readership with implementation details of their software solution. As a researcher who is very interested and involved in parallelization, performance/code/memory optimizations and generally algorithm implementation, I disagree that "it is out of scope of the paper" to present such details in a paper which focuses on the software itself. I think each of those elements should be given its own paragraph or subsection, as it constitutes an important part of the solution. Both on the side of the choice of algorithms, paralellization techniques, data decomposition, and similar. If you claim that something is 1000x times faster than something else, you should back it up with reasons.

Section 4, bullet 5. You should provide HW/SW testing environment and a more clear description on how you tested and compared your solutions to alternatives. Again, 10-1000x requires good explanations.

6. PLOS authors have the option to publish the peer review history of their article (what does this mean?). If published, this will include your full peer review and any attached files.

Reviewer #1: **Yes: **Neetik Mukherjee

Reviewer #2: No

---

## [Author Response · Author response to Decision Letter 1]

17 Dec 2024

We have attached our response as a PDF file answering all reviewer comments.

---

## [Decision Letter · Decision Letter 1]

8 Jan 2025

PONE-D-24-22899R1ComplexityMeasures.jl: scalable software to unify and accelerate entropy and complexity timeseries analysisPLOS ONE

Dear Dr. Datseris,

Thank you for submitting your manuscript to PLOS ONE. After careful consideration, we feel that it has merit but does not fully meet PLOS ONE’s publication criteria as it currently stands. Therefore, we invite you to submit a revised version of the manuscript that addresses the points raised during the review process. **Both reviewers appreciated the replies to their comments, and the amendments to the manuscript. However, one of them raised additional points that can be improved. In particular, the reviewer suggested to stress the main contributions of the work, and to add some details about the software in order to provide a more in-depth description of the tool. I would encourage you to address these final concerns, with the aim of making the paper more valuable for the readers also from a technical viewpoint.**

We look forward to receiving your revised manuscript.

Kind regards,

Alessandro Mengarelli

Academic Editor

PLOS ONE

**Journal Requirements:**

Reviewers' comments:

Reviewer's Responses to Questions

**Comments to the Author**

1. If the authors have adequately addressed your comments raised in a previous round of review and you feel that this manuscript is now acceptable for publication, you may indicate that here to bypass the “Comments to the Author” section, enter your conflict of interest statement in the “Confidential to Editor” section, and submit your "Accept" recommendation.

Reviewer #1: All comments have been addressed

Reviewer #2: (No Response)

2. Is the manuscript technically sound, and do the data support the conclusions?

Reviewer #1: Yes

Reviewer #2: (No Response)

3. Has the statistical analysis been performed appropriately and rigorously? 

Reviewer #1: Yes

Reviewer #2: (No Response)

4. Have the authors made all data underlying the findings in their manuscript fully available?

Reviewer #1: Yes

Reviewer #2: Yes

5. Is the manuscript presented in an intelligible fashion and written in standard English?

Reviewer #1: Yes

Reviewer #2: Yes

6. Review Comments to the Author

**Reviewer #1:** This paper presents ComplexityMeasures.jl, an open-source software for the Julia programming language that implements 1,638 complexity measures in just 3,841 lines of code, demonstrating remarkable efficiency with an average of only 2.3 lines of code per measure. The software's composable design, which emphasizes mathematical rigor, ensures high performance and extendability. ComplexityMeasures.jl offers an extensive and versatile tool set for nonlinear time series analysis, addressing the need for a comprehensive framework to guide researchers in selecting appropriate measures for their specific applications. The paper effectively compares the software with existing alternatives, highlighting its superior computational performance, reliability, and expandability. As part of the DynamicalSystems.jl library, it adheres to open-source principles, fostering a sustainable development community. This contribution is highly valuable and merits immediate acceptance.

**Reviewer #2:** The authors addressed some of my concerns from the previous version, and the paper is clearly better. However, some of them are still not addressed. It might be different perspectives, as they come from the mathematics/statistics department, and I am looking at the paper from the perspective of computer scientist and software engineer. However, the key point of the paper is implemented software, thus relevant details should be given.

Especially confusing, the authors stated in one of the responses that there is no novelty (?!) in their research, except software. The authors should be able to formulate main contributions and novelty of the paper. A piece of software, and in this case, its innovative (composable) design and architecture are surely some contributions to the state of the art in the field of statistics software. So, please, try to emphasize 2-3 main contributions of the paper, bullet-by-bullet, especially from the software perspective.

The other important thing in the revised version of the paper is the repeated lack of design and implementation details and the most important optimization steps taken. There is a glimpse on the design in Section 2.1, however it quickly turns into a user help on how to use information() and complexity() functions. I miss a class diagram of the system, some "behind the scenes" details which make this software performant, not just interface descriptions, although I appreciate design decisions given in lines 150-176.

Although I can understand that majority of target audience are users of the software, this is a research paper with an emphasis on the piece of software, not merely a user software manual. For those from the audience who would like to take the authors' steps and implement a new piece of software or expand the existing one, those descriptions would be valuable, as they give some insights to different challenges and their solutions. So, it is of no interest to track every single optimization, but to emphasize main challenges, problems, and solutions and make for interested reader easier to look at the code.

In the end, it is good to have evidence why something is drastically faster than something else. Reproducibility is more than welcomed in contemporary software solutions and research. However, it is of little interest for the readers to just repeat experiments to confirm that some benchmark is 1000x faster. Evidences should be backed with reasons and explanations. It is not the same when you have ~10x (order of magnitude or less) improvement and ~1000x (three orders of magnitude). The former can be achieved with some implementation optimizations, while the latter are usually the consequence of rather disruptive, novel algorithmic approaches and designs. There are only 5 benchmarks, and it would not be hard to comment 2-3 sentences on each of them in the discusion section.

7. PLOS authors have the option to publish the peer review history of their article (what does this mean?). If published, this will include your full peer review and any attached files.

Reviewer #1: **Yes: **Dr. Neetik Mukherjee

Reviewer #2: No

---

## [Author Response · Author response to Decision Letter 2]

21 Mar 2025

Please see attached text file REPLY_2.txt.

---

## [Decision Letter · Decision Letter 2]

25 Apr 2025

ComplexityMeasures.jl: scalable software to unify and accelerate entropy and complexity timeseries analysis

PONE-D-24-22899R2

Dear Dr. Datseris,

We’re pleased to inform you that your manuscript has been judged scientifically suitable for publication and will be formally accepted for publication once it meets all outstanding technical requirements.

Kind regards,

Alessandro Mengarelli

Academic Editor

PLOS ONE

Additional Editor Comments (optional):

The paper has been properly modified, addressing all the Reviewers comments.

Reviewers' comments:

Reviewer's Responses to Questions

**Comments to the Author**

1. If the authors have adequately addressed your comments raised in a previous round of review and you feel that this manuscript is now acceptable for publication, you may indicate that here to bypass the “Comments to the Author” section, enter your conflict of interest statement in the “Confidential to Editor” section, and submit your "Accept" recommendation.

Reviewer #2: All comments have been addressed

2. Is the manuscript technically sound, and do the data support the conclusions?

Reviewer #2: Yes

3. Has the statistical analysis been performed appropriately and rigorously? 

Reviewer #2: Yes

4. Have the authors made all data underlying the findings in their manuscript fully available?

Reviewer #2: Yes

5. Is the manuscript presented in an intelligible fashion and written in standard English?

Reviewer #2: Yes

6. Review Comments to the Author

Reviewer #2: Thanks to the authors for the effort. I appreciate fine additions to the design decisions, as well as the comments on performance considerations in 4.1. I think the paper is now ready for publication.

7. PLOS authors have the option to publish the peer review history of their article (what does this mean?). If published, this will include your full peer review and any attached files.

Reviewer #2: No

---

## [Editor Report · Acceptance letter]

PONE-D-24-22899R2

PLOS ONE

Dear Dr. Datseris,

I'm pleased to inform you that your manuscript has been deemed suitable for publication in PLOS ONE. Congratulations! Your manuscript is now being handed over to our production team.

Kind regards,

on behalf of

Dr. Alessandro Mengarelli

Academic Editor

PLOS ONE